# Tuning-Free Accountable Intervention for LLM Deployment - A Metacognitive Approach

## Abstract

Large Language Models (LLMs) have catalyzed transformative advances across a spectrum of natural language processing tasks through few-shot or zero-shot prompting, bypassing the need for parameter tuning. While convenient, this modus operandi aggravates "hallucination" concerns, particularly given the enigmatic "black-box" nature behind their gigantic model sizes. Such concerns are exacerbated in high-stakes applications (e.g., healthcare), where unaccountable decision errors can lead to devastating consequences. In contrast, human decision-making relies on nuanced cognitive processes, such as the ability to sense and adaptively correct misjudgments through conceptual understanding. Drawing inspiration from human cognition, we propose an innovative *metacognitive* approach, dubbed **CLEAR**, to equip LLMs with capabilities for self-aware error identification and correction. Our framework facilitates the construction of concept-specific sparse subnetworks that illuminate transparent decision pathways. This provides a novel interface for model *intervention* after deployment. Our intervention offers compelling advantages: (*i*) at deployment or inference time, our metacognitive LLMs can self-consciously identify potential mispredictions with minimum human involvement, (*ii*) the model has the capability to self-correct its errors efficiently, obviating the need for additional tuning, and (*iii*) the rectification procedure is not only self-explanatory but also user-friendly, enhancing the interpretability and accessibility of the model. By integrating these metacognitive features, our approach pioneers a new path toward engendering greater trustworthiness and accountability in the deployment of LLMs.

## 1 Introduction

Natural language processing (NLP) has undergone significant advances in recent years, primarily fueled by the advent of Large Language Models (LLMs) (Raffel et al., 2020; Zhou et al., 2022b; OpenAI, 2023). Despite their laudable achievements, LLMs are not infallible; they err due to factors like "hallucination" (McKenna et al., 2023). These vulnerabilities pose critical challenges for the trustworthy deployment of LLMs in high-stakes settings where errors can precipitate significant repercussions. For example, in the application of LLM-assisted medical diagnoses (Monajatipoor et al., 2022), a single misdiagnosis can inflict profound physical and financial costs on the patient.

Despite its significance, the current literature lacks an effective approach to LLM *intervention* after deployment to help the model overcome those errors. U One intuitive method, *few-shot* or *zero-shot prompting* (Wei et al., 2022; OpenAI, 2023) recently has shown promising results. Users can directly query LLMs and point out their mistakes using usually "hand-crafted" prompts. Though they are simple, the post-prompting performance remains uncertain. Moreover, it necessitates human expertise both for error identification and prompt design. (2) Another potential method is to *fine-tune* part of the parameters in LLMs (*e.g*, the final layers) on erroneously predicted examples (Hardt & Sun, 2023). Besides costly human involvement, this method risks model overfitting on those examples and "catastrophic forgetting" of prior knowledge. (3) Some initial work (Li et al., 2023) repetitively performs *activation-level intervention* on all examples to get better performance, thus resulting in drastically inflated inference latency.

Against this backdrop, we trifurcate the challenges for LLM intervention into three folds. ❶ Firstly, the "black-box" nature of LLMs obscures the malfunction source within the multitude of parameters,

complicating targeted intervention. ❷ Secondly, rectification typically depends on domain experts to identify errors, hindering scalability and automation. ❸ Thirdly, the architectural complexity and sheer size of LLMs render pinpointed intervention an overwhelmingly daunting task.

In this paper, we advocate that an ideal intervention should be *metacognitive*, where LLMs are capable of self-aware error identification and correction. This perspective is informed by several key insights from cognitive science literature: (*a*) **Cognitive Perception of Concepts** - humans demonstrate the ability to swiftly identify and rectify judgment errors by perceptively recognizing essential features, or "concepts" (Malafouris, 2013; Koh et al., 2020). This ability to hone in on vital features underscores the efficiency of human cognitive processes. (*b*) **Neural Sparsity for Efficiency** - building upon the notion of efficiency, the ar-

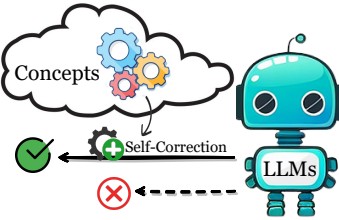

Figure 1: Metacognitive LLMs

chitecture of the human brain provides a valuable lesson. The distribution of neural connections and activity patterns in our brains is characterized by a high degree of sparsity (Gerum et al., 2020). This sparse configuration is believed to facilitate rapid cognitive responses. (*c*) **Conscious Anomaly Detection** - human brain exhibits an intrinsic ability to consciously identify anomalies or challenging problems (Penfield, 2015). Upon encountering such situations, it channels additional neural resources to address them effectively.

Building on this premise, we propose an avant-garde **C**oncept-**L**earning-**E**nabled met**A**cognitive inte**R**vention framework, herein termed **CLEAR**, for LLM deployment. CLEAR facilitates LLMs in mastering concept-specific sparse subnetworks. These subnetworks elucidate transparent decision-making pathways, thereby providing a unique interface for surgical model intervention, that automatically allocates more sparse computing modules to potentially more challenging instances. Distinctively, our approach simultaneously tackles the challenges highlighted above through the following four core contributions:

★ **Metacognition.** At deployment (or inference) time, our metacognitive framework autonomously detects potential mispredictions by measuring logit entropy in pivotal intermediate layers.

★ **Interpretability**. Leveraging the transparency of decision pathways, our **CLEAR** allows for a logical backtrack to the input, thereby aiding user comprehension and fostering trust in the model.

★ **Efficiency.** Upon identification of a misprediction, the LLM architecture dynamically activates extra internal experts to refine concept perception without necessitating further parameter tuning.

★ **Effectivess.** Rigorous experiments on real-world datasets with LLM backbones in various sizes and architectures manifest that our intervention consistently improves inference-time predictions.

## 2 RELATED WORK

**Intervention on Deep Models for Error Mitigation.** Historically, error mitigation in machine learning emphasized simpler models, such as Decision Trees and Random Forests, where corrections were largely heuristic and human-driven (Doshi-Velez & Kim, 2017). With the evolution of machine learning techniques, there was a pivot towards leveraging algorithms themselves for error detection, emphasizing the removal of non-relevant data and unveiling crucial fault-application relationships (Abich et al., 2021). The ascendance of neural networks, and LLMs in particular, brought forth new intervention paradigms. Fine-tuning emerged as a primary strategy for addressing model shortcomings, despite its challenges related to overfitting and catastrophic forgetting of prior knowledge (Wang et al., 2019; French, 1999). Few-shot and Zero-shot prompting marked another avenue, guiding models without altering their internal makeup, leading to inherent limitations in error repeatability (Wei et al., 2022; OpenAI, 2023). Deeper interventions, targeting model architectures, have delivered promising accuracy, yet with computational trade-offs (Li et al., 2023). Notably, quantum error mitigation approaches, though out of our current scope, underline the breadth of exploration in this domain (Subramanian Ravi et al., 2021).

Concurrently, the push towards model interpretability has intensified (Carvalho et al., 2019; Koh et al., 2020; Yuksekgonul et al., 2022). The ultimate goal is to design systems whose inner workings can be easily understood, thereby facilitating targeted interventions. Such transparency is indispensable in critical sectors like healthcare, demanding specialized interventions that are usually hand-carfted by domain experts (Farrell, 2021; Monajatipoor et al., 2022).

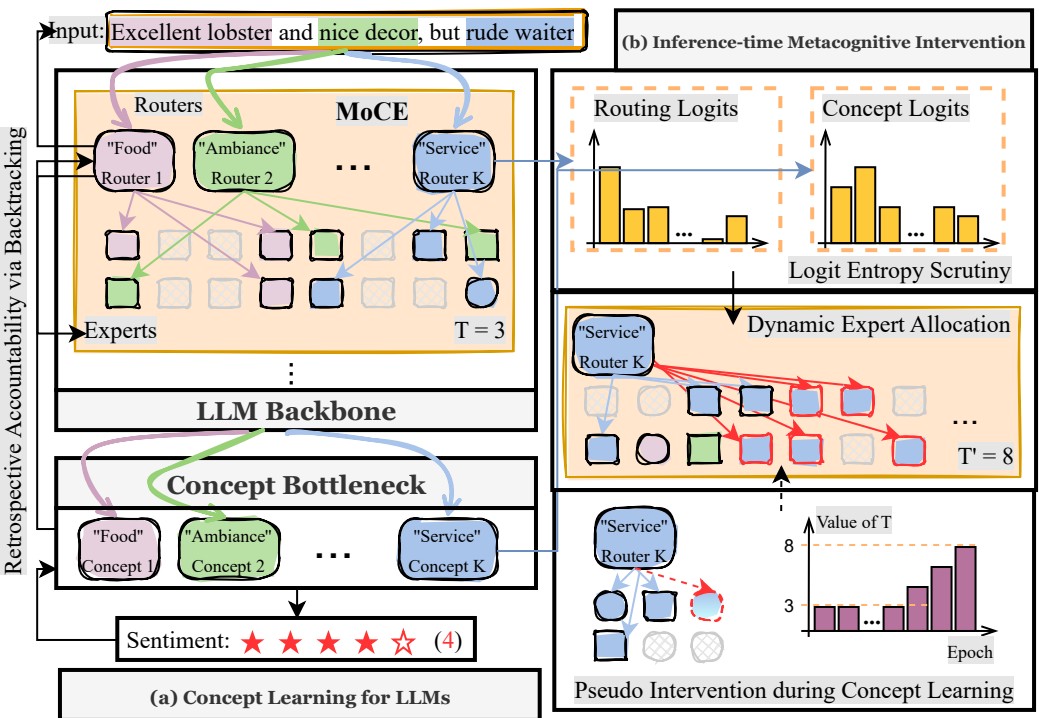

Figure 2: The illustration of the proposed framework **CLEAR**, comprised of two components: (a) *Concept Learinng*, where the LLM backbone learns to construct concept-specific sparse networks via MoCE; and (b) *Metacognitive Intervention*, which involves logit entropy scrutiny, dynamic expert allocation, and pseudo intervention, and offers retrospective accountability.

**Metacognitive Approaches.** Metacognition, eloquently described as "thinking about thinking", has long been acknowledged in cognitive science (Flavell, 1979), resonating through educational and clinical paradigms (Zimmerman, 2013; Moritz & Woodward, 2007). This foundational knowledge has segued into AI, aspiring towards machines with self-reflective and adaptive capabilities (Cox, 2005). Recent endeavors have strived to infuse cognitive inspirations into models, emphasizing a deeper "understanding" of their decisions (Malafouris, 2013). However, genuinely metacognitive LLMs remain an elusive goal, with inherent challenges arising from their black-box nature and their vast, intricate architectures.

## 3 METHODOLOGY

The proposed **C**oncept-**L**earning-**E**nabled met**A**cognitive inte**R**vention framework, **CLEAR** is comprised of two crucial components: (1) *Concept Learning*: the learning of concept-specific sparse subnetworks for LLMs. (2) *Metacognitive Intervention*: automatic error identification and rectification. We provide their detailed elaborations below.

### 3.1 CONCEPT LEARNING FOR LARGE LANGUAGE MODELS

**Basic Setup.** Our primary focus is the enhancement of Large Language Models (LLMs) within the realm of text classification tasks during the inference phase. Given a dataset $\mathcal{D} = \{(\boldsymbol{x}^{(i)}, y^{(i)}, \boldsymbol{c}^{(i)})_{i=1}^{N}\}$, we utilize an LLM, denoted by $f_{\boldsymbol{\theta}}$, to transform an input text $\boldsymbol{x} \in \mathbb{R}^{D}$ into a latent space representation $\boldsymbol{z} \in \mathbb{R}^{E}$. This latent representation is then classified via a linear classifier $g_{\boldsymbol{\phi}}$ into the respective task label $y$. Here $\{\boldsymbol{c}^{(i)}\}_{i=1}^{N}$ denotes the critical features, or "concepts" annotated by humans (Koh et al., 2020; Abraham et al., 2022). These concepts are typically represented using one-hot vectors. Take, for instance, a restaurant review sentiment dataset. The concept "Food" might be denoted by the vector $[0, 0, 1]$, signifying a "Positive" sentiment towards the food. The other vector positions represent alternative sentiments, specifically "Negative" and "Unknown".

**Incorporating Concept Bottlenecks for LLMs.** Our general pipeline is inspired by a previous work (Koh et al., 2020) on image classifications. Instead of altering LLM encoders—which might

compromise the integrity of the text representation—we incorporate a linear layer, characterized by a sigmoid activation function $p_{\psi}$. This layer maps the latent representation $z \in \mathbb{R}^E$ to a concept space $c \in \mathbb{R}^K$, thus creating a decision-making pathway depicted as $x \rightarrow z \rightarrow c \rightarrow y$. By allowing for multi-class concepts, we aim to achieve nuanced interpretations. For ease of reference, LLMs integrated with Concept Bottlenecks are termed LLM-CBMs (e.g., BERT-CBM). The training of LLM-CBMs is dual-faceted: (1) Ensure the concept prediction $\hat{c} = p_{\psi}(f_{\theta}(x))$ aligns with the input's true concept labels $c$. (2) Ensure the label prediction $\hat{y} = g_{\phi}(p_{\psi}(f_{\theta}(x)))$ corresponds with true task labels $y$. Our framework predominantly employs the *joint training* approach due to its superior performance, as supported by Anonymous (2023). The joint training mechanism harmonizes the concept encoder and label predictor through a weighted sum, represented as $\mathcal{L}_{\text{joint}}$:

$$
\begin{aligned}
\theta^*, \psi^*, \phi^* &= \underset{\theta,\psi,\phi}{\arg\min}\, \mathcal{L}_{\text{joint}}(x, c, y) \\
&= \underset{\theta,\psi,\phi}{\arg\min}[\mathcal{L}_{\text{CE}}(g_{\phi}(p_{\psi}(f_{\theta}(x), y) + \gamma \mathcal{L}_{\text{CE}}(p_{\psi}(f_{\theta}(x)), c)] \\
&= \underset{\theta,\psi,\phi}{\arg\min} \sum_{k=1}^{K}[\mathcal{L}_{\text{CE}}(g_{\phi_k}(p_{\psi_k}(f_{\theta}(x), y) + \gamma \mathcal{L}_{\text{CE}}(p_{\psi_k}(f_{\theta}(x)), c_k)],
\end{aligned}
\tag{1}
$$

where, $\mathcal{L}_{\text{CE}}$ represents the Cross-Entropy loss. The third line of the equation incorporates the loss iterating across the concepts, a detail that will prove pivotal soon. Notably, the sensitivity of jointly trained LLM-CBMs to the loss weight $\gamma$ requires attention. By default, we set $\gamma$ to $5.0$, based on its optimized performance as observed in Anonymous (2023). Further details on varying training strategies are expounded in Appendix A. It should be noted that conventional LLM-CBMs (Koh et al., 2020) tend to train all concepts simultaneously. This concurrent training potentially muddles the parameters meant for individual concept prediction, thus hampering precise intervention.

**Building Concept-Specific Sparse Subnetworks via Mixture of Concept Experts.** Our research presents the *Mixture of Concept Experts* (MoCE) framework, a novel approach to creating pathways anchored in specific concepts, thereby enhancing targeted interventions. This model takes cues from mixture-of-expert (MoE) paradigms (Shazeer et al., 2017), known for their dynamic activation of unique network subsets per input. By conditioning on concept-based computation, MoCE crafts sparse modules, fine-tuning the encoding of text inputs as per their inherent concepts.

Conforming to conventions, we structure blocks of MoCEs as the expert layer. This layer comprises a multi-head attention block combined with multiple parallel experts. Specifically, we adapt MoCE for Transformer architectures, integrating MoE layers within successive Transformer blocks. Crafting a MoCE expert typically involves segmenting the conventional MLP of transformers into more compact segments (Zhang et al., 2021) or duplicating the MLP (Fedus et al., 2022). It's noteworthy that the majority of extant MoE studies have predominantly focused on the MLP segment within transformers. This focus arises because MLPs account for approximately two-thirds of the entire model parameter set, serving as key repositories of accrued knowledge within memory networks (Geva et al., 2020; Dai et al., 2022).

The experts can be symbolized as $\{e_m\}_{m=1}^M$, where $m$ signifies the expert index and $M$ is the total count of experts. For each concept $c_k$, an auxiliary routing mechanism, dubbed $r_k(\cdot)$, is deployed. This mechanism identifies the top-$T$ experts based on peak scores $r_k(x)_m$, with $x$ representing the present intermediate input embedding. Generally, $T$ is much smaller than $N$, which underscores the sparse activations among modules of the LLM backbone, making the inference of the model more efficient. The output, $x'$, emanating from the expert layer is:

$$
x' = \sum_{k=1}^{K} \sum_{m=1}^{T} r_k(x)_m \cdot e_m(x); \qquad r_k(x) = \texttt{top-T}(\texttt{softmax}(\zeta(x)), T),
\tag{2}
$$

where $\zeta$ is a shallow MLP representing learnable routers (Fedus et al., 2022). For the $k$th concept, the expert $e_t(\cdot)$ initially processes the given features, after which the router amplifies it using coefficient $r_k(x)_t$. The combined embeddings across concepts yield the output $x'$. The $\texttt{top-T}$ operation retains the top $T$ values, nullifying the others. Typically, a balancing mechanism, such as load or importance balancing loss (Shazeer et al., 2017), is implemented to avert the risk of representation collapse, preventing the system from repetitively selecting the same experts across diverse inputs. Transitioning to matrix representation for all MoE layers in the LLM structure, we derive:

$$
\hat{y} = \sum_{k=1}^{K} \phi_k \cdot \sigma(\psi_k \cdot f_{\theta_k}(x)) = \sum_{k=1}^{K} \phi_k \cdot \sigma(\psi_k \cdot \sum_{m=1}^{T} R_k(x)_m \cdot E_m(x)),
\tag{3}
$$

where $\sigma(\cdot)$ is the sigmoid projector's activation function, with $\boldsymbol{R}(\cdot)$ and $\boldsymbol{E}(\cdot)$ symbolizing matrix incarnations of all expert layer routers and experts. Crucially, equation 3 portrays a factorized decision trajectory, streamlining the classification framework. This can be optimized through a single backward iteration of the composite loss as outlined in equation 2. Note that equation 3 accomplishes a **core objective**: during inference, the LLM backbone's final classifications intrinsically rely on the learned routing policies, the chosen experts, and the perceived concepts. This unique accountability offers an interface for precise error identification and interventions.

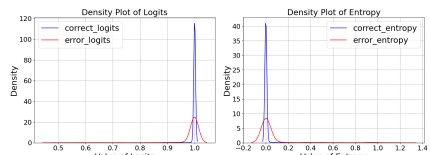

Figure 3: Logit entropy scrutiny. It can be observed that predictions with errors tend to demonstrate lower confidence and larger entropy.

## 3.2 TUNING-FREE METACOGNITIVE INTERVENTION

At its core, our metacognitive intervention emulates human cognitive processes: similar to the way human brains discern potential pitfalls or intricate challenges, our **CLEAR** framework proactively identifies these issues. It then adeptly marshals extra sparse neural resources, specifically experts, to address these challenges. In this subsection, we elucidate how this is realized through our delineated sparse decision pathways, in the form of answering three distinctive research questions.

> **RQ1:** *How to achieve "metacognition" for intervention on LLMs?*
> **A1:** *By autonomously monitoring anomalous pattern at critical intermediate layers.*

▷ *Logit Entropy Scrutiny.* The foremost goal is to automatically identify potential errors or more complex cases. As inferred from Equation equation 3, two critical decision-making phases notably impact the ultimate label prediction: (a) the deduced routing $\{\boldsymbol{R}_k(\boldsymbol{x})\}_{k=1}^K$ of the final MoCE layer, and (b) the determined concept activation $\hat{\boldsymbol{a}} = \{\hat{a}_k\}_{k=1}^K = \boldsymbol{\psi} \cdot f_{\boldsymbol{\theta}}(\boldsymbol{x})$. Intuitively, an elevated entropy of predictive logits denotes a more dispersed distribution over experts or concept options, signifying lower model confidence and pinpointing instances that deserve additional attention. For this purpose, the Shannon entropy is utilized for logits within the routine and concept activation:

$$H(\boldsymbol{p}) = -\sum_{j=1} \texttt{softmax}(l_j)\log(\texttt{softmax}(l_j)). \tag{4}$$

For illustration, the distributions of logits and entropy for concept prediction are depicted using kernel density estimation in Figure 3. It is evident that predictions with errors tend to demonstrate lower confidence and augmented entropy, reinforcing our premise. For automation, as we iterate through the concepts, K-Means clustering is employed to divide confidence levels into two clusters (K=2). The subset with lower confidence is considered to stem from the more challenging instances. K-Means offers the advantage of determining thresholds dynamically, eliminating human involvement. If, for a single concept prediction relating to an instance, the confidence levels of both the routine and concept activation surpass the corresponding thresholds, we tag this concept prediction as potentially erroneous. We demonstrate further study on the scrutiny in Figure 6 (a) and (b).

> **RQ2:** *Once a potential error is identified during inference, how to intervene on LLMs "without extra parameter tuning"?*
> **A2:** *By dynamically allocating experts and enforcing preparatory rehearsal during training.*

▷ *Tuning-free Intervention.* Once an erroneous prediction is identified, we allocate augmented computational resources to secure a more reliable prediction. This operation can be easily achieved by setting the maximum expert number from $T$ to a larger number $T'$ for the router as below. Note that this operation is very efficient since no extra parameter tuning is involved.

$$\boldsymbol{r}_k(\boldsymbol{x}) = \texttt{top-T}(\texttt{softmax}(\zeta(\boldsymbol{x})), T') \tag{5}$$

▷ *Pseudo Intervention during Concept Learning.* Both existing research (Chen et al., 2023) and our experiments (Figure 6 (c) and (d)) indicate that directly adding more experts at the inference stage results in marginal improvements. Drawing inspiration from how humans reinforce understanding of challenging subjects through repeated practice before the final examination, we emulate a similar rehearsal mechanism during concept learning for better metacognitive intervention. As the LLM

model is fine-tuned on the task dataset, we progressively raise the count of experts from $T$ to $T'$ linearly after a predetermined number of training epochs, typically post the halfway mark. This strategy of pseudo intervention during the training phase significantly enhances predictions when the expert count is increased during the inference-time metacognitive intervention, as depicted in Figure 6 (c) and (d). Through this essential rehearsal setup, and by sequentially executing the steps outlined in equation 4 and equation 5, the LLM backbone is empowered to autonomously detect possible errors, addressing them more robustly with minimal human oversight.

> **RQ3:** *How can users understand the intervention?*
> **A3:** *By backtracking from the task label, through the sparse pathway, to the input text.*

▷ *Retrospective Accountability.* A standout feature of our metacognitive intervention is its inherent explicability. Using the decision-making pathways showcased in equation 3, one can trace back from the task label prediction, passing through perceived concepts and activated subnetworks (experts), all the way to the initial text input, as shown in Figure 2. Illustrative examples are provided in Figure 4. The incorporation of our framework, **CLEAR**, represents a harmonious blend of precision, flexibility, and accountability.

## 4 EXPERIMENTS

### 4.1 EXPERIMENTAL SETUP

**Datasets.** Our experiments are conducted on two widely-used real-world datasets: `CEBaB` (Abraham et al., 2022) and `IMDB-C` (Anonymous, 2023). Each of them is a text classification dataset comprised of human-annotated concepts and task labels. Their statistics are presented in Table 1.

Table 1: Statistics of experimented datasets and concepts.

| Dataset | | CEBaB (5-way classification) | | | | IMDB-C (2-way classification) | | |
|---------|-------|-----------|-----------|-----------|-------|-----------|-----------|-----------|
| | **Train / Dev / Test** | | 1755 / 1673 / 1685 | | **Train / Dev / Test** | | 100 / 50 / 50 | |
| **Concept** | **Label** | **Negative** | **Positive** | **Unknown** | **Label** | **Negative** | **Positive** | **Unknown** |
| | Food | 1693 (33.1%) | 2087 (40.8%) | 1333 (26.1%) | Acting | 76 (38%) | 66 (33%) | 58 (29%) |
| | Ambiance | 787 (15.4%) | 994 (19.4%) | 3332 (65.2%) | Storyline | 80 (40%) | 77 (38.5%) | 43 (21.5%) |
| | Service | 1249 (24.4%) | 1397 (27.3%) | 2467 (48.2%) | Emotional Arousal | 74 (37%) | 73 (36.5%) | 53 (26.5%) |
| | Noise | 645 (12.6%) | 442 (8.6%) | 4026 (78.7%) | Cinematography | 118 (59%) | 43 (21.5%) | 39 (19.4%) |

**Baselines.** In this study, our evaluation primarily involves two categories of frameworks as baselines. For an in-depth analysis, we examine both (*a*) the performance on the *test* sets and (*b*) the performance on the *development* sets, before and after the intervention. This dual-faceted examination allows us to assess the intervention's effectiveness and evaluate the model's potential deterioration in generalizability and catastrophic forgetting of critical prior knowledge. Different LLM backbones are employed in our analysis, including BERT (Devlin et al., 2018), OPT (Zhang et al., 2022), and T5 (Raffel et al., 2020). We adjust our choice of LLM backbone per the specific methods employed:

▷ *Direct Intervention Methods*: (*i*) Directly prompting the LLM with human identifying mispredictions. For this method, we use GPT-4 (OpenAI, 2023) as the backbone, as it is widely regarded as the most capable LLM currently. (*ii*) Directly fine-tuning the LLM backbones on mispredicted instances identified by humans. (*iii*) Employing the activation-level intervention method, ITI (Li et al., 2023), mentioned in the introduction.
▷ *Concept Bottleneck Models* (CBMs) support concept-level interventions, but still require human experts to identify mispredictions. We consider the following recent CBM frameworks as baselines: (*iv*) Vanilla CBMs (Koh et al., 2020) map the text into concepts using the LLM backbone and involve another linear classifier to perform the final classification. (*v*) Label-free CBMs (LF-CBMs) (Oikarinen et al., 2022) use GPT-4 to obtain the concept labels. (*vi*) Concept embedding models (CEMs) (Zarlenga et al., 2022) that learn continuous embeddings for concepts.

### 4.2 SUPERIOR PERFORMANCE OF CLEAR

The comparative results are presented in Table 2. Reported scores are the averages of three independent runs. Our work is based on general text classification implementations. The implementation of

Table 2: We compare model performance on the `CEBaB` and `IMDB-C` datasets, using *Macro F1* as the evaluation metric, expressed in percentages (%). Scores shaded in gray highlight instances where the model experienced catastrophic forgetting, leading to a decline in performance on the development set. Scores shaded in red indicate a decrease in performance following the intervention. Scores shaded in blue are from CLEAR.

| Methods | Backbones | CEBaB | | | | | | | | IMDB-C | | | | | | | |
|---|---|---|---|---|---|---|---|---|---|---|---|---|---|---|---|---|---|
| | | Pre-intervention | | | | Post-intervention | | | | Pre-intervention | | | | Post-intervention | | | |
| | | Dev | | Test | | Dev | | Test | | Dev | | Test | | Dev | | Test | |
| | | Concept | Task | Concept | Task | Concept | Task | Concept | Task | Concept | Task | Concept | Task | Concept | Task | Concept | Task |
| *Direct Intervention Methods* | | | | | | | | | | | | | | | | | |
| Prompting | GPT4 | - | 46.52 | - | 45.87 | - | 46.52 | - | 48.32 | - | 69.35 | - | 68.74 | - | 69.35 | - | 69.84 |
| Fine-tuning | BERT | - | 80.03 | - | 79.75 | - | 76.43 | - | 81.23 | - | 74.52 | - | 72.11 | - | 71.69 | - | 74.26 |
| | OPT | - | 82.65 | - | 81.37 | - | 80.84 | - | 82.16 | - | 80.62 | - | 79.98 | - | 75.42 | - | 81.05 |
| | T5 | - | 82.64 | - | 82.65 | - | 80.67 | - | 83.34 | - | 81.85 | - | 79.87 | - | 77.62 | - | 81.53 |
| ITI | T5 | - | 82.64 | - | 82.65 | - | 82.64 | | 83.29 | - | 81.85 | - | 79.87 | - | 81.85 | - | 81.25 |
| *Concept Bottleneck Models* | | | | | | | | | | | | | | | | | |
| Vanilla-CBMs | BERT | 85.86 | 78.32 | 85.29 | 78.11 | 85.86 | 78.32 | 88.52 | 79.52 | 64.52 | 72.51 | 62.76 | 70.41 | 64.52 | 72.51 | 65.31 | 71.96 |
| | OPT | 87.84 | 80.03 | 87.27 | 79.73 | 87.84 | 80.03 | 89.62 | 80.12 | 67.15 | 78.96 | 66.53 | 78.21 | 67.15 | 78.96 | 69.47 | 79.34 |
| | T5 | 88.20 | 81.05 | 87.96 | 80.63 | 88.20 | 81.05 | 90.21 | 81.05 | 68.85 | 79.58 | 67.94 | 78.26 | 68.85 | 79.58 | 70.26 | 79.95 |
| LF-CBMs | BERT | 82.37 | 75.24 | 83.45 | 75.69 | 82.37 | 75.24 | 83.52 | 75.82 | 62.51 | 70.49 | 60.35 | 68.21 | 62.51 | 70.49 | 61.32 | 68.13 |
| | OPT | 84.54 | 77.62 | 84.62 | 76.84 | 84.54 | 77.62 | 85.36 | 76.64 | 64.18 | 75.24 | 63.37 | 75.06 | 64.18 | 75.24 | 63.58 | 74.65 |
| | T5 | 85.68 | 78.25 | 85.74 | 77.22 | 85.68 | 78.25 | 85.59 | 76.87 | 65.16 | 76.83 | 64.92 | 76.30 | 65.16 | 76.83 | 64.43 | 75.68 |
| CEMs | BERT | 86.78 | 79.10 | 86.62 | 78.64 | 86.78 | 79.10 | 88.67 | 80.04 | 64.86 | 72.61 | 62.84 | 71.05 | 64.86 | 72.61 | 65.57 | 72.33 |
| | OPT | 87.98 | 80.51 | 87.92 | 79.86 | 87.98 | 80.51 | 89.89 | 80.65 | 68.29 | 79.67 | 66.97 | 78.68 | 67.84 | 79.62 | 70.34 | 79.75 |
| | T5 | 88.64 | 81.32 | 88.34 | 80.69 | 88.64 | 81.32 | 90.65 | 81.42 | 68.98 | 79.83 | 68.65 | 79.64 | 68.98 | 79.83 | 70.93 | 80.72 |
| *Metacognition Intervention* | | | | | | | | | | | | | | | | | |
| **CLEAR** | OPT-MoCE | 88.24 | 80.96 | 88.24 | 80.39 | 89.04 | 80.85 | 90.46 | 81.24 | 68.83 | 79.75 | 68.47 | 79.52 | 68.39 | 79.86 | 71.02 | 80.12 |
| **CLEAR** | T5-MoCE | 89.65 | 81.62 | 89.63 | 81.30 | 89.65 | 81.62 | 91.25 | 82.14 | 69.46 | 80.25 | 69.65 | 80.63 | 69.46 | 80.25 | 71.67 | 80.95 |

our framework is also released[1]. More implementation details and parameter values are included in Appendix B. From the result, we obtain the following findings:

- **Effectiveness.** The presented framework, CLEAR, unfailingly surpasses all baseline models in concept prediction and task label prediction, both before and after the intervention. This consistent outperformance underscores the robustness and efficiency of the CLEAR framework across various conditions and parameters. (*a*) In the concept learning phase, the proposed MoCE layers play a pivotal role. By constructing sparse, concept-specific subnetworks, the MoCE layers facilitate the efficient disentanglement of concepts. This organized division significantly smoothens and enhances the internalization of concepts, laying a solid foundation for further enhancement during the intervention phase. (*b*) During the intervention phase, the excellence of CLEAR further shines. It elevates prediction accuracy through precisely targeted interventions, tailoring its approach to the specific challenges and complexities encountered in each instance. This meticulous and adaptable strategy allows CLEAR to hone in on and address the unique difficulties faced by each prediction task, ensuring optimal enhancement of prediction accuracy.

- **Metacognition**. Beyond raw performance metrics, the CLEAR framework profoundly underscores its metacognitive prowess, presenting a triumvirate of decisive advantages: *efficiency*, *accountability*, and *autonomy*, setting it distinctly apart from existing baselines. (*a*) *Efficiency:* Unlike direct intervention methods, CLEAR is free from extensive tuning, safeguarding it from prevalent issues like catastrophic forgetting encountered in fine-tuning methods (shaded in gray). (*b*) *Autonomy:* Distinct from CBMs, CLEAR operates without human intervention, ensuring complete autonomy. This self-sufficiency expands its applicability, particularly in areas where human expertise is limited or costly. Notably, LF-CBMs, utilizing GPT-4 to extract noisy concept labels, display a detrimental effect from intervention (highlighted in pink). This observation further underscores the criticality of accurate and targeted intervention. (*c*) *Accountability:* CLEAR provides a comprehensive, multilayered insight into its decision-making process, covering concept, subnetwork, and input levels. This transparency significantly amplifies user trust, offering clarity and assurance in the framework's operations and decisions. We will go through more details of those advantages in subsequent subsections.

## 4.3 EXTRA INVESTIGATION AND ABLATION STUDY

**Accoutability.** CLEAR does not just execute tasks; it stands out by ensuring retrospective interpretability and in-depth insight into its metacognitive intervention processes. This transparency permeates various levels through backtracking, offering concept-level, subnetwork-level, and input-level explanations. This multilayered insight not only fulfills intellectual curiosity but also enhances user trust and confidence in CLEAR. By understanding the "how" and "why" behind each decision, users gain a more profound insight into the model's operations, leading to informed and confident interaction with the framework.

---

[1] https://github.com/Anonymous-submit-23/CLEAR.git

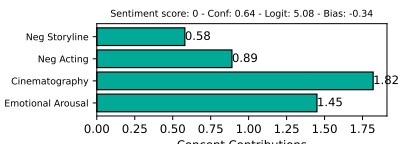

... The story is not that good... The acting is pretty poor too... manges to work with some very creepy visuals and atmosphere... It's a quiet kind of horror that isn't made anymore...

| | Emotional Arousal | Acting | Storyline | Cinemato-graphy | Task Label |
|---|---|---|---|---|---|
| $c_k/y$ | + | - | - | + | + |
| $\hat{c}_k/\hat{y}$ | + | - | - | - | - |
| $\hat{c}'_k/\hat{y}'$ | + | - | - | + | + |

Figure 4: Illustration of an case study for the accountable metacognitive intervention.

▷ *Case Study.* To further illustrate, we present a detailed case study of the metacognitive intervention process in Figure 4. More examples are included in Appendix C. This depiction illuminates the transition of the predicted label for the concept "Cinematography" from incorrect "-" to correct "+", subsequently refining the final task label.

Sentiment score: 0 - Conf: 0.64 - Logit: 5.08 - Bias: -0.34

Neg Storyline: 0.58
Neg Acting: 0.89
Cinematography: 1.82
Emotional Arousal: 1.45

Concept Contributions

Figure 5: Contributions of Concepts.

Texts highlighted in red indicates the clues overlooked by insufficient experts. Moreover, by analyzing expert and concept activations before and after the intervention, we reveal the neural mechanics underpinning the intervention strategy at the subnetwork level, offering additional real-world implications. For instance, we can compute the influence $I$ of each concept $c_k$ to the final decision by the product of the concept activation $\hat{a}_k$ and the corresponding weight $w_k$ in the linear classifier: $I(c_k) = \hat{a}_k \cdot w_k$. The results are visualized in Figure 5. This capability to correct and interpret the underlying causes for prediction errors further boosts the model's overall trustworthiness and usability. Coupled with experimental findings, this case study enriches our understanding of the potential of the proposed metacognitive interventions, showcasing them not just as a model fine-tuning method, but as a structured approach towards more transparent and adaptable AI systems.

**Autonomy and Efficiency.** CLEAR also demonstrate unique advanatges with its full autonomy and tuning-free interventions. We list the comparison of important features among all intervention methods in Table 3. From the comparison, we can observe that CLEAR is the only framework that achieves this impressive enhancement without the need for extensive human involvement or intricate parameter tuning, which are often required by other existing methods. This self-sufficient functionality not only streamlines the operation of the CLEAR framework but also reinforces its reliability and effectiveness. The absence of heavy reliance on human input or complex tuning procedures eliminates potential sources of error and inconsistency, further bolstering the robustness, consistency and dependability of the CLEAR framework.

Table 3: Efficiency comparison between interventions

| Method | Human labels | Parameter tuning | Targeted intervention |
|---|---|---|---|
| Prompting | ✔ | ✘ | ✘ |
| Fine-tuning | ✔ | ✔ | ✘ |
| ITI | ✘ | ✘ | ✘ |
| CBM | ✔ | ✘ | ✘ |
| CLEAR | ✘ | ✘ | ✔ |

Table 4: Ablation study on intervention mechanism. Scores are reported in %.

| Methods | CEBaB | | | | | | IMDB-C | | | | | |
|---|---|---|---|---|---|---|---|---|---|---|---|---|
| | Pre-intervention | | Post-intervention | | Improvement (↑) | | Pre-intervention | | Post-intervention | | Improvement (↑) | |
| | Concept | Task | Concept | Task | Concept | Task | Concept | Task | Concept | Task | Concept | Task |
| CLEAR (oracle) | 89.63 | 81.30 | 91.98 | 82.06 | 2.35 | 0.76 | 69.65 | 80.63 | 72.64 | 81.36 | 2.99 | 0.73 |
| CLEAR | 89.63 | 81.30 | 91.25 | 81.80 | 1.62 | 0.5 | 69.65 | 80.63 | 71.67 | 80.95 | 2.02 | 0.32 |

**Ablation Study.** In this section, we perform comprehensive ablation studies to evaluate the critical components of CLEAR, including the *intervention mechanism* options, *logit entropy scrutiny*, and *pseudo intervention*. We will discuss each result in detail.

▷ *Intervention Mechanism.* In Table 4, we present a detailed comparison between the proposed metacognitive intervention and oracle intervention. For the oracle intervention, human-annotated ground-truth labels serve as the oracle, ensuring all incorrect predictions are identified. This method allows for the precise allocation of additional experts to these accurately identified mispredictions during the intervention phase. Analyzing the results, it is evident that CLEAR performs commendably, only marginally lagging behind the oracle intervention. This close performance highlights the robust metacognitive capabilities of CLEAR. Despite not having access to human-annotated labels as the oracle method does, CLEAR effectively identifies and corrects erroneous predictions with a high degree of accuracy. This successful outcome underscores the efficiency and reliability of CLEAR's metacognitive intervention mechanisms, demonstrating its practical utility and effectiveness in real-world applications where human annotation may not always be feasible or accurate.

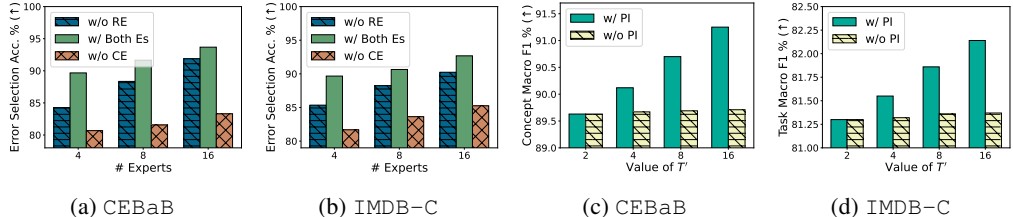

| (a) CEBaB | (b) IMDB-C | (c) CEBaB | (d) IMDB-C |

Figure 6: Extra studies about CLEAR. (a) and (b) investigate diverse logit entropies for scrutiny under different expert numbers, where RE denotes *routing entropy*, and CE denotes *concept prediction entropy*. (c) and (d) examine the effects of w/wo *pseudo intervention* (PI) on gradually increased intervention expert number $T'$.

▷ *Options for Logit Entropy Scrutiny.* Figure 6 (a) and (b) visually lay out the outcomes for various logit entropy scrutiny methods. These visualizations offer an examination into the effectiveness of different scrutiny methods. Analytically, it is unmistakably observed that employing both entropy thresholds jointly contributes to superior performance compared to the utilization of each individually. This synergy between the thresholds manifests as a more robust and resilient model, able to more accurately navigate and correct its predictions. Specifically, the exclusion of concept prediction entropy results in a marked decline in performance. This downturn is attributed to the distinctive structure of CLEAR, which constructs concept-specific subnetworks. This architecture is more sensitive to concept prediction errors, and awareness of these errors is pivotal for the model's functionality. Recognizing and addressing these errors directly enhances the capacity for accurate and effective intervention. It allows the model to pinpoint and rectify the specific areas of miscalculation, bolstering the overall performance and reliability of CLEAR.

▷ *Pseudo Intervention.* Figure 6 (c) and (d) illustrate the performance difference of CLEAR with and without the proposed pseudo intervention during concept learning. The results clearly demonstrate that employing pseudo intervention significantly enhances CLEAR's performance. This positive outcome confirms our premise that intentionally increasing the number of experts during training better prepares the model for inference-time intervention, leading to improved results. The pseudo intervention acts as a robust rehearsal, honing the model's capabilities and reinforcing its readiness for real-time challenges, thereby affirming its crucial role in the CLEAR framework.

▷ *Sensitivity Analysis on the Number of Experts.* Figure 6 (a) and (b) distinctly emphasize the notable enhancement in CLEAR's performance as the number of experts in the MoCE layers is amplified (larger model parameters). This remarkable advancement is fundamentally due to the natural expansion of the model, leading to a consequential augmentation in its learning capability. A more intricate network of experts within the layers allows for a more comprehensive learning phase, enabling the model to make more accurate and refined predictions and decisions. Conversely, Figure 6 (c) and (d) underscore the significant improvement in CLEAR's performance when more experts are engaged in correcting erroneous predictions during the intervention phase. This data corroborates the vital role of a higher number of experts in both the learning and intervention stages of the model, showcasing their contribution to the superior performance of CLEAR.

## 5   CONCLUSION

In conclusion, CLEAR stands out as a pioneering framework, uniquely positioned to alleviate the contemporary challenges faced by Large Language Models (LLMs). This paper outlines its robust capabilities in autonomously identifying and correcting errors, thereby reducing the need for extensive human oversight and intricate adjustments. By employing a metacognitive strategy inspired by human cognitive processes, CLEAR enables the construction of transparent, concept-specific sparse subnetworks. This attribute ensures clear, comprehensible decision pathways and eases post-deployment model intervention. In tackling the enduring "black-box" issue prevalent in LLMs, CLEAR confidently showcases its effectiveness in diminishing mispredictions and bolstering overall model interpretability and accessibility. These advances by CLEAR underscore a significant enhancement in both the performance and reliability of LLMs, ensuring their more trustworthy and accountable deployment in diverse real-world scenarios. Moving forward, the widespread application of CLEAR promises a tangible, positive shift in the landscape of LLM deployment, underscoring its role as an invaluable asset in the evolution of machine learning models.

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

# A   DEFINITIONS OF DIFFERENT TRAINING STRATEGIES

Given a text input $x \in \mathbb{R}^D$, concepts $c \in \mathbb{R}^K$ and its label $y$, the strategies for fine-tuning the text encoder $f_\theta$, the projector $p_\psi$ and the label predictor $g_\phi$ are defined as follows:

*i) Vanilla fine-tuning an LLM:* The concept labels are ignored, and then the text encoder $f_\theta$ and the label predictor $g_\phi$ are fine-tuned either as follows:

$$\theta, \phi = \arg\min_{\theta,\phi} \mathcal{L}_{CE}(g_\phi(f_\theta(x), y),$$

or as follows (frozen text encoder $f_\theta$):

$$\phi = \arg\min_{\phi} \mathcal{L}_{CE}(g_\phi(f_\theta(x), y),$$

where $\mathcal{L}_{CE}$ indicates the cross-entropy loss. In this work we only consider the former option for its significant better performance.

*ii) Independently training LLM with the concept and task labels:* The text encoder $f_\theta$, the projector $p_\psi$ and the label predictor $g_\phi$ are trained seperately with ground truth concepts labels and task labels as follows:

$$\theta, \psi = \arg\min_{\theta,\psi} \mathcal{L}_{CE}(p_\psi(f_\theta(x)), c),$$

$$\phi = \arg\min_{\phi} \mathcal{L}_{CE}(g_\phi(c), y).$$

During inference, the label predictor will use the output from the projector rather than the ground-truth concepts.

*iii) Sequentilally training LLM with the concept and task labels:* We first learn the concept encoder as the independent training strategy above, and then use its output to train the label predictor:

$$\phi = \arg\min_{\phi} \mathcal{L}_{CE}(g_\phi(p_\psi(f_\theta(x)), y).$$

*iv) Jointly training LLM with the concept and task labels:* Learn the concept encoder and label predictor via a weighted sum $\mathcal{L}_{joint}$ of the two objectives described above:

$$\theta, \psi, \phi = \arg\min_{\theta,\psi,\phi} \mathcal{L}_{joint}(x, c, y)$$
$$= \arg\min_{\theta,\psi,\phi} [\mathcal{L}_{CE}(g_\phi(p_\psi(f_\theta(x)), y)$$
$$+ \gamma \mathcal{L}_{CE}(p_\psi(f_\theta(x)), c)].$$

It's worth noting that the LLM-CBMs trained jointly are sensitive to the loss weight $\gamma$. We tune the value for $\gamma$ for better performance (Anonymous, 2023).

# B   IMPLEMENTATION DETAIL

In this section, we provide more details on the implementation settings of our experiments. Specifically, we implement our framework with PyTorch (Paszke et al., 2017) and HuggingFace (Wolf et al., 2020) and train our framework on a single 80 GB Nvidia A100 GPU. We follow a prior work (Abraham et al., 2022) for backbone implementation. All backbone models have a maximum token number of 512 and a batch size of 8. We use the Adam optimizer to update the backbone, projector, and label predictor according to Section 3.1. The values of other hyperparameters (Table 5 in the next page) for each specific PLM type are determined through grid search. We run all the experiments on 4 Nvidia A100 GPUs with 80GB RAM.

For the LLM backbones, we use their pubic versions available on `Huggingface`. Specifically, we deploy `bert-base-uncased`, `facebook/opt-350m`, and `t5-base`. In our implementation, we also include other baseline backbones from more languae model families. We intentionally include the above three in the main experiment results for their similar sizes. The other backbones include: `roberta-base`, `distilbert-base-uncased`, `gpt2`, `facebook/opt-125m`, `facebook/opt-1.3b`, and `switch-transformer-base`.

Table 5: Key parameters in this paper with their annotations and evaluated values. Note that **bold** values indicate the optimal ones.

| Notations | Specification | Definitions or Descriptions | Values |
|---|---|---|---|
| max_len | - | maximum token number of input | 128 / 256 / **512** |
| batch_size | - | batch size | 8 |
| epoch | - | maximum training epochs | 30 |
| lr | DistilBERT | learning rate when the backbone is DistilBERT | 1e-3 / 1e-4 / **1e-5** / 1e-6 |
| | BERT | learning rate when the backbone is BERT | 1e-3 / 1e-4 / **1e-5** / 1e-6 |
| | RoBERT | learning rate when the backbone is RoBERT | 1e-3 / 1e-4 / **1e-5** / 1e-6 |
| | OPT-125M | learning rate when the backbone is OPT-125M | 1e-3 / 1e-4 / **1e-5** / 1e-6 |
| | OPT-350M | learning rate when the backbone is OPT-350 | 1e-4 / 1e-5 / **1e-6** / 1e-7 |
| | OPT-1.3B | learning rate when the backbone is OPT-1.3B | 1e-4 / 1e-5 / **1e-6** / 1e-7 |
| | CLEAR | learning rate for CLEAR | 1e-4 / **3e-4** / 5e-4 / 7e-4/ 1e-5 |
| $\gamma$ | DistilBERT | value of $\gamma$ when the backbone is DistilBERT | 1 / 3 / **5** / 7 / 9 |
| | BERT | value of $\gamma$ when the backbone is BERT | 1 / 3 / **5** / 7 / 9 |
| | RoBERT | value of $\gamma$ when the backbone is RoBERT | 1 / 3 / **5** / 7 / 9 |
| | OPT-125M | value of $\gamma$ when the backbone is OPT-125M | 1 / 3 / **5** / 7 / 9 |
| | OPT-350M | value of $\gamma$ when the backbone is OPT-350 | 1 / 3 / **5** / 7 / 9 |
| | OPT-1.3B | value of $\gamma$ when the backbone is OPT-1.3B | 1 / 3 / 5 / **7** / 9 |
| | CLEAR | value of $\gamma$ for CLEAR | 5 / 7 / 9 / **10** / 11 / 13 / 15 |

## C  MORE EXAMPLES FROM REAL-WORLD DATASETS

Figure 7: An example for the metacognitive intervention on one instance from the `CEBaB` dataset.

Figure 8: An example for the metacognitive intervention on one instance from the `IMDB-C` dataset.

## D  COMPARISON WITH EXISTING WORKS ON MOE FOR LLMS

**Mixture of Experts in Large Language Models.** The incorporation of Mixture of Experts (MoE) into Large Language Models (LLMs) has evolved significantly, with early research by Shazeer et al. (2017) laying the groundwork. These foundational studies (Fedus et al., 2022; Zhou et al., 2022a; Du et al., 2022; Artetxe et al., 2021; Shen et al., 2023) focused primarily on improving model performance and computational efficiency in a black-box manner. On the contrary, in this work, we utilize the design of MoE in LLMs for metacognitive capabilities. This novel approach, distinct from earlier efficiency-focused applications, uses MoE for error detection and correction, a critical step towards solving the interpretability and trust issues in AI decision-making. Our framework, CLEAR, contributes to this evolving landscape by embedding MoE within a metacognitive framework, emphasizing error rectification, transparency, and autonomy in LLMs. This shift marks a significant advancement from traditional MoE applications, positioning CLEAR at the forefront of innovative LLM enhancement strategies.

# E   ANALYSIS OF K-MEANS IN LOGITS SCRUTINY

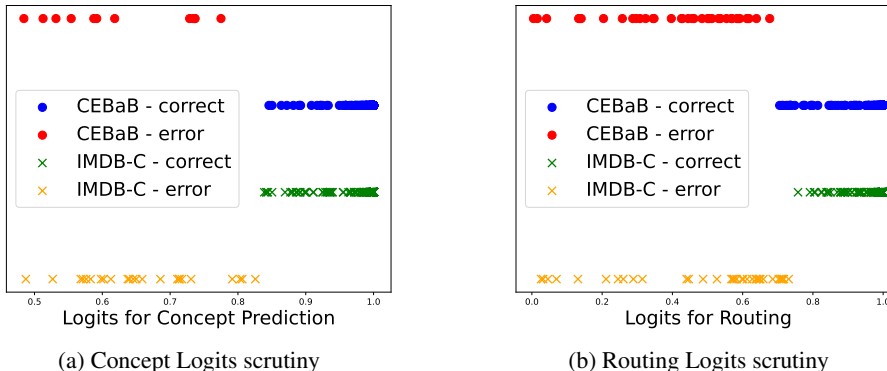

(a) Concept Logits scrutiny        (b) Routing Logits scrutiny

Figure 9: Studies on the design of K-means for logits scrutiny. This figure illustrates the effectiveness of K-means in distinguishing between correct and erroneous logits for both routing and concept prediction. Logits are normalized using a softmax function, reducing the impact of noise and extreme values.

# F   ANALYSIS OF OVERFITTING IN CONCEPT LEARNING

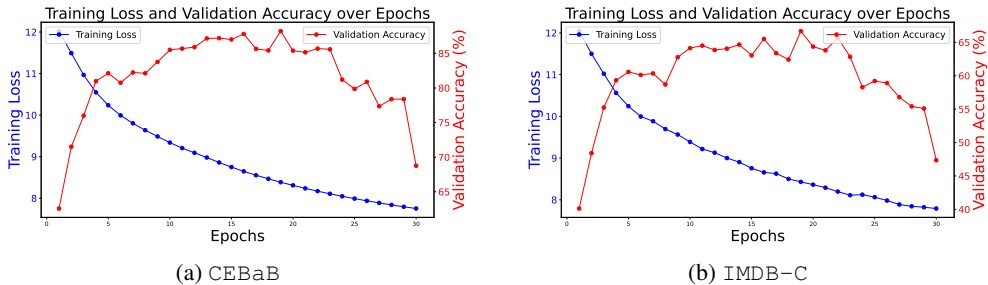

(a) CEBaB             (b) IMDB-C

Figure 10: Visualization of training dynamics of one run on CEBaB and IMDB-C datasets. We adopt the "early stop" strategy to avoid overfitting, where the model with the highest validation accuracy is selected and evaluated on the test set.

# G   COMPUTATION COMPLEXITY AGAINST NUMBER OF EXPERTS

In Figure 11, we show an experiment to quantify the increase in computational complexity, as measured by FLOPs, with the increasing number of experts, in one the inference given a sequence of 128 tokens. The results, as shown in the attached figure, clearly indicate a linear increase in computational complexity with the number of experts. This is an expected outcome, as each additional expert adds a similar computational load to the model. However, it is crucial to note that the addition of each expert provides a significant improvement in model performance. We acknowledge that there is a trade-off between the increased computational overhead and the enhanced performance achieved. In practical applications,

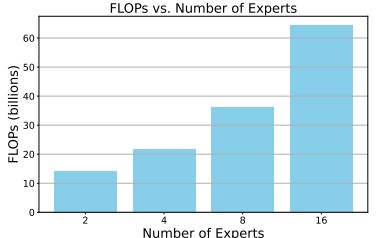

Figure 11: The number of floating-point multiplication operations (FLOPs) consumed on one the inference with a sequence of 128 tokens.

this trade-off would need to be carefully balanced based on the specific requirements and constraints of the task at hand. In future work, we plan to investigate techniques for optimizing the computational efficiency of our model, such as pruning less effective experts or implementing more efficient computation strategies. This will allow us to maintain or even enhance model performance while managing the computational costs more effectively.

