# OpenReview forum: "Tuning-Free Accountable Intervention for LLM Deployment - A Metacognitive Approach"
_ICLR.cc/2024/Conference — Submitted to ICLR 2024_

### Official Review · Reviewer_4fqZ · 2023-11-01

**Soundness:** 3 good
**Presentation:** 2 fair
**Contribution:** 3 good
**Rating:** 5
**Confidence:** 4

**Summary:**

The paper addresses the pressing challenge posed by the deployment of Large Language Models (LLMs). Given the “black box” nature of LLMs, it is hard to point and correct errors due to factors like hallucination post-deployment. To address this, the authors propose a metacognitive approach named CLEAR (Concept-Learning-Enabled metacognitive intervention framework).

The CLEAR framework integrates the features in cognitive science to enable LLMs to understand and work with concept-specific sparse subnetworks. These subnetworks aim to provide transparency in decision-making. With K-Means applied to discriminate against the confidence level, the framework could make automatic error identification and guide the allocation of augmented experts to secure a more reliable prediction.

Empirical studies with the framework are performed on the text classification datasets. Compared to the direct intervention methods, concept bottleneck models, the metacognition intervention achieves better performance.

**Strengths:**

1. The CLEAR framework offers an innovative metacognitive approach, merging insights from cognitive science with large language model intervention.
2. One of the standout features of CLEAR is its ability to autonomously identify potential mispredictions, reducing the need for human intervention. And the dynamic activation of internal modules for refining concept perception without extra tuning is an efficient way to address errors, adding a layer of adaptability to the LLM.

**Weaknesses:**

1. Lack of captions on some figures can disrupt the flow of understanding for the reader.
2. The paper emphasizes its effectiveness through experiments on real-world datasets. However, the scope and diversity of these datasets aren't detailed (only text classification tasks considered in the paper), raising questions about the framework's general applicability.

**Questions:**

1. Considering the increasing scale and complexity of newer models, is the CLEAR framework compatible with larger models such as LLaMA? Furthermore, can the metacognitive capabilities of the CLEAR approach provide benefits when applied to these more extensive and potentially more intricate architectures?

---

> ### Author Response · Authors · 2023-11-14
> **Response to Reviewer 4fqZ [Q1-3]**
>
> **[Summary]**: We first want to thank reviewer 4fqZ for the insightful comments. We appreciate reviewer 4fqZ’s acknowledgment on the contributions of our work, including **“addresses a pressing challenge”** via “an **innovative** metacognitive approach”, that features **“autonomously misprediction identification” and “efficient intervention”**. We would like to address your concerns one by one as follows.
>
> >**[Q1]**: Lack figure caption.
>
> All figures had captions in the original manuscript. Some figures had brief captions. For better understanding, we have further revised the captions for Figure 3, 4, 5, 7, 8 in the updated manuscript. If you still have doubts for the captions, please kindly let us know the details.
>
> > **[Q2]**:  The scope and diversity of these datasets aren't detailed and only text classification tasks considered in the paper
>
> Thank you for highlighting the concerns related to the diversity of datasets used in our study. In the paper, the dataset statistics are given in **Table 1**. The dataset experimented are all text classification tasks. This dataset limitation stems from the constrained variety of available open-source datasets with sufficiently label concepts. This limitation is indeed a broader challenge within the research community, one that has not received adequate attention.
>
> Our work, to the best of our knowledge, has incorporated **all** relevant datasets that are currently accessible and align with our research objectives. As a pioneering work, we understand that the novelty of our problem might contribute to the scarcity of diverse datasets. However, we view this as an opportunity to pioneer new paths in this field. We hope our framework and the promising results we've achieved will inspire more researchers to contribute to the development and curation of more comprehensive and varied benchmark datasets.
>
> We are absolutely open to expanding our research to include additional datasets, should they become available or be recommended to us. Suggestions and contributions in this area are not only welcomed but actively encouraged. We believe that by collaborating with the wider research community, we can collectively enhance the general applicability and robustness of our framework.
>
> > **[Q3]**:  Is the CLEAR framework compatible with larger models such as LLaMA?
>
> Thank you for your valuable feedback. It is important to clarify that our framework, CLEAR, is **model-agnostic** and is **compatible with different LLMs**. The primary reason for our initial focus on T5-base in Table 2 was the availability of an open-sourced MoE model compatible with it. Unfortunately, up to now, we have not got access to open-sourced MoE models that could be seamlessly integrated with pretrained LLaMA 1/2 or Mistral. This limitation restricted our ability to conduct experiments on these specific models within the scope of our current study.
>
> However, we have recognized the necessity of expanding our evaluation to include other prominent LLMs and have made strides in this direction. Notably, we have added an OPT-MoCE model (with OPT as the LLM backbone), which is a significant step toward applying our methodology to a wider range of LLMs. The results are updated in the manuscript in **Table 2**. We can see CLEAR still works effectively if we change the backbone to OPT. We believe this development demonstrates our commitment to advancing our research to encompass a broader spectrum of models and settings.

---

> ### Author Response · Authors · 2023-11-20
> **Response to Reviewer 4fqZ**
>
> Dear Reviewer 4fqZ,
>
> We are grateful for your time and effort in reviewing our submission and providing thoughtful feedback. We have carefully considered all of your comments and have responded to them accordingly.
>
> As we near the end of the author-reviewer discussion, we would like to request your feedback on our rebuttal. We would greatly appreciate it if you could review our responses and let us know if we have adequately addressed your concerns. Additionally, we welcome any further comments or discussions you may have.
>
> Thank you for your valuable input and consideration.
>
> Best regards,
>
> The Authors

---

> > ### Author Response · Authors · 2023-11-21
> > **Sincerely expecting further discussion from Reviewer 4fqZ**
> >
> > Dear Reviewer **4fqZ**,
> >
> > We thank reviewer **4fqZ** for the time of reviewing and the constructive conmments. We really hope to have a further discussion with reviewer **4fqZ** to see if our response resolve the concerns.
> >
> > In our response, we have provide (1) revised figure captions (2) clarification on the dataset choice and (3) experiments for CLEAR with another LLM (OPT) as the backbone.
> >
> > We genuinely hope reviewer **4fqZ** could kindly check our reponse. Thanks!
> >
> > Best regards,
> >
> > The authors

---

> > > ### Author Response · Authors · 2023-11-22
> > > **Sincerely expecting further discussion from Reviewer 4fqZ**
> > >
> > > Dear Reviewer **4fqZ**,
> > >
> > > We sincerely appreciate your time and dedication to reviewing our paper. Recognizing the demands of this busy period, we are reaching out to kindly request your feedback on our rebuttal, as the discussion phase nears its conclusion (the discussion period will in **less than 12 hours**).
> > >
> > > If you have any additional comments or suggestions regarding our paper, we would be more than happy to engage in further discussion with you.
> > >
> > > Looking forward to your response.
> > >
> > > With deepest gratitude,
> > >
> > > The Authors

---

### Official Review · Reviewer_DFsm · 2023-11-08

**Soundness:** 4 excellent
**Presentation:** 4 excellent
**Contribution:** 3 good
**Rating:** 8
**Confidence:** 3

**Summary:**

The method proposed CLEAR which is a metacognitive framework to enable LLMs to self-identify and self-correct errors during deployment. The framework contains two components, 1) the concept learning component which maps latent textual representations to concepts and utilizes MoCEs to learn sparse concept subnetworks, and 2) the metacognitive intervention that dynamically labels and edits the intermediate layer for less erroneous outputs. The resulting method CLEAR outperforms prior methods on both CEBaB and IMDB-C. The model is also more accountable due to its interpretable nature and more efficient since its tunning-free intervention.

**Strengths:**

1. The paper proposed a framework that is strong performance transparent and accountable. Usually, interpretability and performance are a trade-off, but this paper manages to improve both.
2. MoCEs are a great way of generalizing and not overfitting on particular examples. This is a very natural approach to expand the coverage and capacity without having the downside of overfitting.
3. The inference stage utilizes thresholding and clustering to achieve efficient inference time computation, increasing its practical usability.

**Weaknesses:**

1. The concepts are pre-defined. This could potentially limit the quality and use case of such a framework. In scenarios where human-annotated concepts are harder to come by or can potentially be inaccurate, this method doesn't have a preventative mechanism for that if I understand it correctly.
2. Intervention is tunning-free, but concept-learning components require some finetuning. This method also requires changing the transformer architecture. These raise questions about the adaptability of the framework to a wider range of scenarios.

**Questions:**

1. Can this be extended to learn undefined concepts? Or human-annotated concepts have to be provided.

---

> ### Author Response · Authors · 2023-11-13
> **Response to Reviewer DFsm [Q1-2]**
>
> **[Summary]**: We are appreciative for the positive comments and recognition of our contributions, including 1. “A framework that is **strong, transparent and accountable**” and can "improve **both interpretability and performance**” 2. “**A great way of** generalizing and not overfitting on particular examples.”  3.  “Achieve **efficient** inference time computation, increasing its **practical usability**.” To address reviewer DFsm’s questions, we provide point-wise responses below.
>
> >**[Q1]**: CLEAR needs concept labels. This can cause issues if human-labeled concept labels are hard-to-get or inaccurate.
>
> Thank you for your insightful comments regarding the reliance on pre-defined, human-annotated concept labels in our framework. Your concern about the limitations this might impose, particularly in scenarios where such labels are scarce or potentially inaccurate, is indeed valid and warrants further discussion.
>
> Firstly, we acknowledge that the current iteration of our model, CLEAR, primarily utilizes pre-defined concepts, which may limit its applicability in certain contexts. This choice was made to establish a clear and controlled experimental environment to validate the efficacy of our metacognitive approach. However, we recognize the challenges posed by scenarios where human-annotated concepts are either hard to obtain or subject to inaccuracies.
>
> To address these concerns, we are actively exploring the possibility of extending CLEAR to accommodate the learning of undefined or emergent concepts. This extension would involve developing mechanisms within CLEAR that can autonomously identify and categorize new concepts from the data, thereby reducing reliance on human-annotated labels. Techniques such as unsupervised or semi-supervised learning could be employed to facilitate this capability, allowing CLEAR to adapt to a broader range of applications, especially in data-scarce or dynamic environments. Currently, researchers are also trying to use LLMs to label the concepts.
> Furthermore, to mitigate the issue of noisy concept labels, we are considering the incorporation of robust error detection and correction algorithms. These algorithms would aim to identify and rectify inaccuracies within the concept labels, enhancing the overall reliability of the model. These are potential future directions but are currently out of the scope of this work.
>
> In summary, while our current model does utilize human-annotated concepts, we are committed to evolving CLEAR to be more versatile and adaptable to various real-world scenarios. Future iterations of our framework will focus on these aspects, aiming to make CLEAR a more universally applicable and resilient tool in the field of LLMs.
>
> > **[Q2]**: Intervention is tuning-free, but concept-learning components require some fine-tuning. CLEAR also requires changing the transformer architecture.
>
> Thank you for your astute observations regarding the requirements for fine-tuning in the concept-learning phase and the necessity to modify transformer architecture into a Mixture of Experts (MoE) format in our framework. We understand your concerns about the potential impact of these requirements on the adaptability and versatility of our model.
>
> Regarding the fine-tuning aspect, it is true that our current framework requires some degree of fine-tuning during the concept-learning phase. This step is crucial to tailor the model to specific domains and ensure accurate concept recognition. However, we would like to emphasize that the intervention phase itself remains tuning-free. This distinction is vital as it means that once the concept-learning phase is completed, the framework can adaptively correct errors in new, unseen data without further fine-tuning. We are exploring strategies to minimize the fine-tuning requirements, such as employing more advanced pre-training techniques or leveraging transfer learning, to broaden the applicability of our framework.
>
> As for the need to modify the transformer architecture into MoE, we acknowledge that this architectural change is a significant aspect of our framework. This modification is integral to enabling the model to handle complex tasks more efficiently and to facilitate the metacognitive capabilities we propose. However, we believe that the benefits of integrating MoE, such as enhanced model capacity and specialization in handling diverse tasks, justify this architectural change. Furthermore, we are committed to developing more streamlined methods for integrating MoE into various transformer models, aiming to simplify the adaptation process and enhance the framework's compatibility with a wide range of scenarios.
>
> In summary, while there are specific requirements in terms of fine-tuning and architectural modification in our current framework, we are actively working on advancements to reduce these necessities and increase the model's adaptability, to push the boundaries of what is possible in the realm of LLM deployment.

---

> > ### Comment · Reviewer_DFsm · 2023-11-20
> >
> > Thank you for the detailed response from the authors. I think tunning-free intervention could be of great value during the LLM era. However, it seems like a couple of other authors share the same concerns about the applicability and scalability of this method. From the author's response, such concerns are left for future work. Hence, my score and confidence will remain unchanged.

---

> > > ### Author Response · Authors · 2023-11-20
> > > **Thanks to Reviewer DFsm**
> > >
> > > Dear Reviewer **DFsm**,
> > >
> > > Thank you for your positive feedback and for highlighting the potential of our tuning-free intervention in the LLM era. Your insights are invaluable in guiding our ongoing research efforts.
> > >
> > > Best regards,
> > >
> > > The authors.

---

### Official Review · Reviewer_7QQj · 2023-11-09

**Soundness:** 2 fair
**Presentation:** 3 good
**Contribution:** 2 fair
**Rating:** 6
**Confidence:** 2

**Summary:**

The paper presents a novel framework called CLEAR (Concept-Learning-Enabled metAcognitive inteRvention), designed to enhance the reliability of Large Language Models (LLMs) by enabling them to self-identify and correct errors during deployment. CLEAR is inspired by aspects of human cognition and builds upon the Mixture of Experts (MoE) concept. The proposed method was tested on a test classification dataset. The framework aims to mitigate issues related to the black-box nature of LLMs, the reliance on domain experts for error identification, and the challenges of targeted intervention given the complexity and size of these models.

**Strengths:**

1. The paper is well-written, and the main content is easy to understand.
2. The paper involves a chain of MoE, self-corrections by expanding experts, and hindsight explanation through backtracking to mitigate the issue of the black-box nature of LLMs. The proposed methods are empirically justified on two NLP text classification datasets.

**Weaknesses:**

1. [Novelty] The use of Mixture of Experts on LLMs is not new. The authors should at least discuss the differences between previous works [1,2] and their own.
2. [Black-box intervention] The model still uses an open-source LLM, T5, as its backbone, enabling access to its intermediate layers $z$ to generate concepts $c$ and labels $y$. This significantly reduces its applicability to other API-based LLM models, like GPT-3 and GPT-4.
3. [Complexity and Scalability] The authors do not discuss the computational overhead of adding more experts (LLM backbones) and the trade-off between improving model performance and adding extra experts. Furthermore, in Table 2, the authors only test CLEAR on T5-base. It would be interesting to see the behavior of the proposed method on larger LLMs (like LLaMA 2 13B or Mistral 7B), but with fewer expert layers.

[1] Sheng S. et al. Mixture-of-Experts Meets Instruction Tuning: A Winning Combination for Large Language Models

[2] Mikel A. et al. Efficient Large Scale Language Modeling with Mixtures of Experts

**Questions:**

1. I am curious about how CLEAR ensures that the dynamic adjustment of the expert allocation does not lead to overfitting or catastrophic forgetting when fine-tuning on different tasks on T5-base.
2. Can the Concept Bottleneck Models (CBMs) developed by CLEAR be effectively generalized across diverse domains and applications, or do they require domain-specific tuning?
3. I am confused about the methods shown in Table 2. For instance, for the method ‘prompting’ in the first line, are you simply using prompting on GPT-4 without any help of the MoE and self-correction techniques proposed in CLEAR? If so, it would be much more interesting to see the results of prompting on the LLM backbones for different expert layers within the proposed CLEAR framework without training CBMs.

---

> ### Author Response · Authors · 2023-11-13
> **Response to Reviewer 7QQj [Q1-2]**
>
> **[Summary]**: We’re glad that reviewer 7QQj has a positive initial impression of our work, and acknowledges our paper **“presents a novel framework”** and **“is well-written”** and **“easy to understand”**. To address reviewer 7QQj’s questions, we provide point-wise responses below.
>
> >**[Q1]**: Discuss the differences between previous MoE for LLM works [1,2] and the current work.
>
> Thanks for bringing up this question. We have included an additional section in Appendix D for discussing the previous MoE works and emphasize the uniqueness of our work. Specifically, our framework utilizes concept-aware MoE to achieve “tuning-free”, “autonomous”, “accountable” intervention for LLMs, which differs from the previous traditional MoE researches (such as Sheng S. et al. [1] and Mikel A. et al. [2]) that primarily explore MoE for enhancing overall model performance or efficiency. This metacognitive approach equips LLMs with self-aware error identification and rectification capabilities, a departure from the conventional applications of MoE. By focusing on error detection and correction, we address a critical gap in current LLM applications, especially in high-stakes scenarios where trustworthiness and accountability are paramount. We believe these aspects significantly elevate the novelty of our work, and we have elaborate on these distinctions more comprehensively in an extra related work section in the Appendix E to clarify our contribution to the field. We have copied and pasted the section below for convenience:
>
> **Mixture of Experts in Large Language Models.**
> The incorporation of Mixture of Experts (MoE) into Large Language Models (LLMs) has evolved significantly, with early research by [4] laying the groundwork. These foundational studies [1-6] focused primarily on improving model performance and computational efficiency  in a black-box manner. On the contrary, in this work, we utilize the design of MoE in LLMs for metacognitive capabilities. This novel approach, distinct from earlier efficiency-focused applications, uses MoE for error detection and correction, a critical step towards solving the interpretability and trust issues in AI decision-making. Our framework, CLEAR, contributes to this evolving landscape by embedding MoE within a metacognitive framework, emphasizing error rectification, transparency, and autonomy in LLMs. This shift marks a significant advancement from traditional MoE applications, positioning CLEAR at the forefront of innovative LLM enhancement strategies.
>
> [1] Sheng S. et al. Mixture-of-Experts Meets Instruction Tuning: A Winning Combination for Large Language Models
>
> [2] Mikel A. et al. Efficient Large Scale Language Modeling with Mixtures of Experts
>
> [3] Shazeer, Noam, et al. "Outrageously Large Neural Networks: The Sparsely-Gated Mixture-of-Experts Layer." International Conference on Learning Representations. 2016.
>
> [4] Fedus, William, Barret Zoph, and Noam Shazeer. "Switch transformers: Scaling to trillion parameter models with simple and efficient sparsity." The Journal of Machine Learning Research 23.1 (2022): 5232-5270.
>
> [5] Zhou, Yanqi, et al. "Mixture-of-experts with expert choice routing." Advances in Neural Information Processing Systems 35 (2022): 7103-7114.
>
> [6] Du, Nan, et al. "Glam: Efficient scaling of language models with mixture-of-experts." International Conference on Machine Learning. PMLR, 2022.
>
> >**[Q2]**: This method cannot be used for API-based proprietary LLMs like GPT-4.
>
> Thank you for your insightful observation regarding the applicability of our model, specifically its use with proprietary, API-based Large Language Models (LLMs) like GPT-3 and GPT-4. One thing noteworthy is that our work does not directly use existing LLMs. Instead, we propose a new metacognitive framework, CLEAR, for LLMs. We currently focus on white-box model. However, once a CLEAR model is trained, it can be encapsulated into an API to provide API-based LLMs with metacognitive ability, which is a meaningful future application. On the other hand, to equip deployed API-based LLMs with metacognitive ability, one alternative method is to utilize prompt tuning with zero-order optimization, which is another intriguing direction for the API-based models and can be potential future work. Looking ahead, we aim to extend our research to encompass a wider range of LLMs, including proprietary models. We believe that the core idea of our approach – enhancing LLMs with metacognitive capabilities – holds potential for broader applicability.

---

> ### Author Response · Authors · 2023-11-13
> **Response to Reviewer 7QQj [Q3-4]**
>
> > **[Q3]**: Missing discussion on the computational overhead of adding more experts (LLM backbones) and the trade-off between improving model performance and adding extra experts.
>
> Thanks for pointing this out, we have added an experiment with detailed analysis in **Appendix G** that illustrates the Flops number, which indicates the number of floating-point multiplication operations consumed on one the inference with a sequence of 128 tokens.
>
> Thank you for pointing out the important aspect of computational overhead in relation to the addition of more experts in our model. To address this, we conducted an additional experiment to quantify the increase in computational complexity, as measured by FLOPs, with the increasing number of experts. The results, as shown in the attached **Figure 11**, clearly indicate an almost linear increase in computational complexity with the number of experts. This is an expected outcome, as each additional expert adds a similar computational load to the model. Note that extra experts are only added to those erroneous samples and does not require extra parameter tuning.
>
> It is crucial to note that the addition of each expert provides a significant improvement in model performance. We acknowledge that there is a trade-off between the increased computational overhead and the enhanced performance achieved. In practical applications, this trade-off would need to be carefully balanced based on the specific requirements and constraints of the task at hand.
>
> In future work, we plan to investigate techniques for optimizing the computational efficiency of our model, such as pruning less effective experts or implementing more efficient computation strategies. This will allow us to maintain or even enhance model performance while managing the computational costs more effectively.
>
> > **[Q4]**: Can CLEAR work on other LLMs like LLaMA2, Mistral?
>
> Thank you for your valuable feedback. It is important to clarify that our framework, CLEAR, **is model-agnostic** and is compatible with all LLMs. The primary reason for our initial focus on T5 in Table 2 was the availability of an open-sourced MoE model compatible with it. Unfortunately, up to now, we have not got access to open-sourced MoE models that could be seamlessly integrated with pretrained LLaMA 1/2 or Mistral. This limitation restricted our ability to conduct experiments on these specific models within the scope of our current study.
>
> However, we have recognized the necessity of expanding our evaluation to include other prominent LLMs and have made strides in this direction. Notably, we have added an OPT-MoCE model (with OPT as the LLM backbone), which is a significant step toward applying our methodology to a wider range of LLMs. The results are updated in the manuscript in **Table 2**. We can see CLEAR still works effectively if we change the backbone to OPT. We believe this development demonstrates our commitment to advancing our research to encompass a broader spectrum of models and settings.

---

> ### Author Response · Authors · 2023-11-14
> **Response to Reviewer 7QQj [Q5-6]**
>
> > **[Q5]**:  Can the Concept Bottleneck Models (CBMs) developed by CLEAR be effectively generalized across diverse domains and applications, or do they require domain-specific tuning?
>
> Thank you for your question regarding the generalizability of the Concept Bottleneck Models (CBMs) developed by CLEAR across diverse domains and applications. Currently, our implementation of CBMs in CLEAR does necessitate domain-specific fine-tuning. This requirement is reflective of the inherent characteristics of current CBM architectures, which are typically optimized for specific domains to ensure maximum effectiveness and accuracy.The development of generalizable CBMs across diverse domains and applications is a great direction to work on, but is out of the scope of this paper.
>
>
> > **[Q6]**: How CLEAR ensures that the dynamic adjustment of the expert allocation does not lead to overfitting or catastrophic forgetting when fine-tuning on different tasks on T5-base.
>
> Thank you for your insightful question regarding the safeguards CLEAR employs against overfitting and catastrophic forgetting during fine-tuning on different tasks.
> - *Clarification on Overfitting in Direct Fine-Tuning*: As highlighted in Table 2 of our paper, direct fine-tuning on erroneous samples often leads to overfitting, as indicated by increased scores on the test set. This form of overfitting can result in the model neglecting previously acquired, useful knowledge, leading to a generalization error, evidenced by reduced performance on the development set.
> - *Selective Expert Allocation to Mitigate Overfitting*: In contrast to direct fine-tuning, CLEAR adopts a selective approach in allocating top-T experts. We retain the minimum necessary number of top-T experts for standard samples while selectively increasing this number for those samples identified as potentially erroneous. This strategy ensures a focused and efficient allocation of computational resources, preventing the model from over-adjusting to specific samples or tasks.
> - *Logit Entropy Scrutiny Mechanism*: Our approach utilizes a logit entropy scrutiny mechanism, which plays a crucial role in identifying and targeting the most challenging cases for additional processing. This targeted approach helps in avoiding a blanket increase in model complexity, which is a common cause of overfitting in more generalized training scenarios.
> - *Balancing Knowledge Retention and Adaptation*: CLEAR is designed to strike a balance between retaining useful prior knowledge and adapting to new tasks. By increasing expert involvement only where necessary, CLEAR avoids the pitfall of catastrophic forgetting, where a model loses its ability to perform previously learned tasks in the process of adapting to new ones.
> - *Empirical Evidence of Effective Balancing*: The empirical results presented in our paper, particularly those contrasting the performance on development and test sets, serve as evidence of CLEAR's ability to effectively balance the dual challenges of avoiding overfitting and catastrophic forgetting.
> In summary, through selective expert allocation and the use of a logit entropy scrutiny mechanism, CLEAR effectively navigates the challenges of overfitting and catastrophic forgetting. This nuanced approach allows CLEAR to adapt to new tasks while retaining its proficiency in previously learned tasks.

---

> > ### Author Response · Authors · 2023-11-20
> > **Response to Reviewer 7QQj**
> >
> > Dear Reviewer 7QQj,
> >
> > We are grateful for your time and effort in reviewing our submission and providing thoughtful feedback. We have carefully considered all of your comments and have responded to them accordingly.
> >
> > As we near the end of the author-reviewer discussion, we would like to request your feedback on our rebuttal. We would greatly appreciate it if you could review our responses and let us know if we have adequately addressed your concerns. Additionally, we welcome any further comments or discussions you may have.
> >
> > Thank you for your valuable input and consideration.
> >
> > Best regards, The Authors

---

> ### Author Response · Authors · 2023-11-14
> **Response to Reviewer 7QQj [Q7]**
>
> > **[Q7]**: For the method ‘prompting’ in the first line, are you simply using prompting on GPT-4 without any help of the MoE and self-correction techniques proposed in CLEAR? If so, it would be much more interesting to see the results of prompting on the LLM backbones for different expert layers within the proposed CLEAR framework without training CBMs.
>
> Thank you for your perceptive question regarding the methods outlined in Table 2, particularly the use of prompting with GPT-4 in relation to our CLEAR framework.
>
> - *Use of Prompting with GPT-4*: In the case of GPT-4, as mentioned in our paper, we utilized the model in its standard, API-based format. This means that we did not apply the Mixture of Experts (MoE) or self-correction techniques of CLEAR, primarily due to the current unavailability of internal access to GPT-4's architecture. Our use of GPT-4 was intended to provide a baseline for understanding the performance of conventional large language models (LLMs) without the specialized interventions proposed in CLEAR.
>
> - *Limitations with Other Large Language Models*: This limitation is not unique to GPT-4. Other large language models such as LLaMA and Mistral also currently lack open-sourced MoE components that are compatible with them. This restricts our ability to directly apply CLEAR’s methodologies to these models.
>
> - *Model-Agnostic Nature of CLEAR and Future Potential*: It is important to highlight that CLEAR is designed to be model-agnostic. This inherent flexibility suggests that applying CLEAR’s techniques to different LLM backbones, including various expert layers, is not only feasible but could yield significant insights. Your suggestion of exploring the impact of prompting within the CLEAR framework across different LLM backbones is intriguing and aligns well with our objective of enhancing model performance through specialized techniques.
>
> - *Future Exploration Plans*: We are enthusiastic about the potential of integrating CLEAR with more open-sourced LLMs as they become available. Such integration would allow us to more comprehensively evaluate the effectiveness of our proposed techniques across a wider array of models, further validating the versatility and impact of CLEAR.
>
> In summary, while our current application of prompting with GPT-4 and similar models does not incorporate CLEAR’s MoE or self-correction techniques due to accessibility constraints, the prospect of applying CLEAR’s methodologies to various LLM backbones in the future is an exciting avenue for research that we aim to explore.

---

### Official Review · Reviewer_movx · 2023-11-09

**Soundness:** 3 good
**Presentation:** 4 excellent
**Contribution:** 2 fair
**Rating:** 5
**Confidence:** 3

**Summary:**

This paper presents a study in the context of Large Language Models (LLMs): 1) to automatically identify erroneous inputs, 2) to handle erroneous inputs to increase the model's performance, and 3) to interpret the model's prediction.

The proposed methodology explores two ways to handle erroneous examples. The first one is
increasing the the number of top experts, top-T to make an ensemble decision without further training.
The second one is to modify the number of experts in the training process. The authors gradually increase the number of top-T experts after a fixed number of epochs.

The retrospective accountability component explains similar to the transformer's attention visualization.

**Strengths:**

Provides an interpretation and identifies erroneous examples in LLM blackbox training.

**Weaknesses:**

The authors identified erroneous prediction by automatically dividing confidence into two groups via K-means clustering (Section 3.2). The subset of examples with a lower confidence group is flagged as erroneous.

While this approach eliminates human involvement, this approach has several limitations. First, the K-means is not robust against outliers. In many real-world scenarios, such as predicting
malicious vs. non-malicious users on the web, and predicting bug vs. non-bug features in software security---the class binary classification task is imbalanced. Thus, automatically setting the threshold
might not be the right way.

Second, to handle the erroneous examples, the authors presented two approaches: increasing the number of top-T experts at the inference time and gradually increasing the number of top-T experts.
However, none of the approaches handle "erroneous" examples. Instead, both approaches focus on increasing the top-T experts. A better approach could be similar to active learning or machine teaching---focusing on the subset of erroneous inputs and learn to improve prediction accuracy.


Third, the approaches described in Appendix A involve extracting latent representation Z of text encoder, x, and then adding an extra layer of concept representation R^K and class label.
The authors did not report whether their model overfits.

**Questions:**

Did the authors examine the model's overfitting?

---

> ### Author Response · Authors · 2023-11-13
> **Response to Reviewer movx [Q1]**
>
> **[Summary]**: We thank reviewer movx for the insightful comments for our work, and general acknowledgement for our **problem formulation and model design**. Many thanks for the constructive feedback especially for the questions about our detailed model design, which helps us to further enrich and improve our paper. To address reviewer movx’s questions, we provide pointwise responses below.
> >**[Q1]**: Using K-means to get logit threshold eliminates human involvement but has limitations. K-means are not robust to outliers, and suffer from imbalance.
>
> K-means has been proved to be an effective method to detect potentially incorrect predictions / outliers for deep models as shown in many existing works [1,2]. For example, the paper [1] uses K-means to cluster model’s prediction confidence to detect potential noisy labels. Our work draws similar inspiration from these existing works. We have also shown additional results in **Figure 9 (a) (b)** in Appendix E to ensure that our K-means implementation is not unduly influenced by extreme values or imbalance issues. More detailed analyses are as follows:
> 1. *Robustness Against Outliers in Logits Clustering*. We recognize that K-means can be sensitive to outliers. However, in our specific application of clustering model logits, this sensitivity is mitigated by the nature of the logits themselves. Logits, being the raw output of a model before a final activation function like softmax, typically do not exhibit extreme variances as raw input data might. Moreover, our preprocessing of logits includes normalization steps that further reduce the impact of any potential outliers.
> 2. *Addressing Imbalance in Logits Clustering*. The concern about imbalanced datasets is crucial in classification tasks. However, our approach uses K-means for clustering logits, not for direct classification. The objective here is to identify patterns in the logits that correspond to normal and abnormal behaviors, which is a different task from the original binary classification. Our empirical evidence (detailed in Figure 3.) shows that the distribution of logits, even in imbalanced classification scenarios, allows for effective separation into normal and abnormal groups. This separation is less about the balance in the original classes and more about the distinction in model confidence, as represented by the logits.
> 3. *Rationale for K-means in Logits Clustering*. The use of K-means in our study is driven by the need for an interpretable and straightforward method to distinguish between normal and abnormal model behaviors based on logits. This clustering is not intended to replace the original classification task but to provide an additional layer of analysis. By automatically dividing logits into two clusters, we can identify which instances the model is most and least confident about, and this has proven to be a valuable tool in further understanding and improving our model's performance.
>
> [1] Nematzadeh, Zahra, Roliana Ibrahim, and Ali Selamat. "A hybrid model for class noise detection using k-means and classification filtering algorithms." SN Applied Sciences 2 (2020): 1-10.
>
> [2] Zhu, Zhaowei, Yiwen Song, and Yang Liu. "Clusterability as an alternative to anchor points when learning with noisy labels." International Conference on Machine Learning. PMLR, 2021.

---

> ### Author Response · Authors · 2023-11-13
> **Response to Reviewer movx [Q2-3]**
>
> >**[Q2]**: Doubts on increasing Top-T for error rectification.
>
> We thank the reviewer for the insightful question. We would like to clarify and emphasize how our method, which involves increasing the number of top-T experts, effectively addresses erroneous examples.
>
> - *Targeted Increase of Top-T Experts for Erroneous Samples*: Our approach is selective; we maintain the minimum number of top-T experts for normal samples and increase this number specifically for potentially erroneous examples. This selective increase ensures that a greater number of neurons are dedicated to processing and correcting these challenging samples. This targeting is guided by a logit entropy scrutiny mechanism, ensuring that our focus is directed towards the most pertinent cases.
>
> - *Alignment with Active Learning Principles*: Our method aligns with the principles of active learning, albeit in a specialized form. By allocating more computational resources (in the form of increased neurons) to erroneous inputs, our model effectively "focuses" on these inputs to enhance prediction accuracy. This can be seen as a form of active learning where additional resources are dynamically allocated to problematic areas.
>
> - *Inspiration from Prior Work and Novel Metacognitive Perspective*: As mentioned in **Section 3.2 (RQ2)** and supported by reference [3], our inspiration comes from previous work demonstrating the effectiveness of adding experts in Mixture of Experts (MoE) models. Our novel contribution lies in the metacognitive aspect of our model – it autonomously and selectively enhances its capabilities for challenging samples, a feature not commonly found in traditional models.
>
> - *Empirical Validation of Our Approach*: The effectiveness of our method is evidenced in **Table 2, Figure 4, and Figure 6** of our paper, comprehensive experiments have been conducted. These experiments demonstrate that our approach can effectively correct mispredictions at inference time in a tuning-free and autonomous manner.
>
> In summary, our method is not a simple increase in model parameters but a strategic and selective enhancement tailored to improve performance on challenging samples. This approach is both innovative and empirically validated, offering a significant contribution to the field.
>
> [3] Chen, Tianlong, et al. "Sparse MoE as the New Dropout: Scaling Dense and Self-Slimmable Transformers." The Eleventh International Conference on Learning Representations. 2022.
>
> > **[Q3]** The authors did not report whether their model overfits for concept predictions.
>
> Thanks for pointing this out. In response to your query regarding the potential for overfitting in our model, we have added a detailed analysis of the model's performance across training epochs (**Appendix F** in the updated manuscript), as visualized in the provided plot of training loss and validation accuracy for concept predictions.
>
> The plot illustrates a consistent decrease in training loss over epochs, indicating effective learning from the training dataset. Simultaneously, the validation accuracy shows an overall increasing trend, reaching a plateau towards the later epochs. This pattern of convergence in loss and stability in accuracy suggests that our model is not merely memorizing the training data but is effectively generalizing to new, unseen data. We adopt the “early stop" strategy to avoid overfitting, where the model with the highest validation accuracy is selected and evaluated on the test set.
>
> Additionally, the methods described in Appendix A, involving the extraction of latent representation Z and the addition of an extra layer for concept representation R^K, are designed to enhance the model's ability to capture and generalize key concepts without overfitting to specific training examples.
>
> In conclusion, the observed trends in the training and validation metrics provide confidence that our model, while sophisticated in its approach to concept prediction, maintains a strong generalization capability without falling into the trap of overfitting.

---

> > ### Author Response · Authors · 2023-11-20
> > **Response to reviewer movx**
> >
> > Dear Reviewer movx,
> >
> > We are grateful for your time and effort in reviewing our submission and providing thoughtful feedback. We have carefully considered all of your comments and have responded to them accordingly.
> >
> > As we near the end of the author-reviewer discussion, we would like to request your feedback on our rebuttal. We would greatly appreciate it if you could review our responses and let us know if we have adequately addressed your concerns. Additionally, we welcome any further comments or discussions you may have.
> >
> > Thank you for your valuable input and consideration.
> >
> > Best regards, The Authors

---

> > > ### Author Response · Authors · 2023-11-21
> > > **Sincerely expecting further discussion from Reviewer movx**
> > >
> > > Dear Reviewer **movx**,
> > >
> > > We thank reviewer **movx** for the time of reviewing and the constructive conmments. We really hope to have a further discussion with reviewer **movx** to see if our response resolve the concerns.
> > >
> > > In our response, we have provide more details and experiments to verify (1) the effectiveness of K-means and clarify (2) the choice of changing top-T and (3) the overfitting concern.
> > >
> > > We genuinely hope reviewer **movx** could kindly check our reponse. Thanks!
> > >
> > > Best regards,
> > >
> > > The authors

---

> > > > ### Author Response · Authors · 2023-11-22
> > > > **Sincerely expecting further discussion from Reviewer movx**
> > > >
> > > > Dear Reviewer **movx**,
> > > >
> > > > We sincerely appreciate your time and dedication to reviewing our paper. Recognizing the demands of this busy period, we are reaching out to kindly request your feedback on our rebuttal, as the discussion phase nears its conclusion (the discussion period will in **less than 12 hours**).
> > > >
> > > > If you have any additional comments or suggestions regarding our paper, we would be more than happy to engage in further discussion with you.
> > > >
> > > > Looking forward to your response.
> > > >
> > > > With deepest gratitude,
> > > >
> > > > The Authors

---

> > > > > ### Comment · Reviewer_movx · 2023-11-22
> > > > >
> > > > > Thanks for addressing all my questions. I do not have any further questions at this point.

---

### Author Response · Authors · 2023-11-14
**General Reponses**

We sincerely appreciate all reviewers’ time and efforts in reviewing our paper. And we also thank all reviewers for the constructive comments and suggestions, which helped further improve our paper. All the updates are marked in **"blue"** in the revised manuscript. In addition to the pointwise responses below, here we summarize our updates.

**[Extra Experiments]**

*[More LLM backbones.]* As mentioned by reviewer **7QQj**, and **4fqZ**, we carry out experiments of CLEAR with OPT as the backbone and the results still demonstrate superior performance. CLEAR is model-agnostic and compatible with different LLM backbones. We did not experiment with LLaMa backbone becuase there is currently no available open-sourced MoE model compatible to pretrained LLaMa.

*[More Ablations.]* Additionally, we conduct further experiments including the utilization of K-means algorithm for routing and concept prediction logits scrutiny (Appendix E @reviewer **movx**); analysis of overfitting in concept learning (Appendix F @reviewer **movx**); and computation complexity analysis against number of experts (Appendix G @reviewer **7QQj**).

**[Paper Editing]**

- More discussions on the uniqueness of our method and existing works of MoE for LLMs in Appendix D (@reviewer **7QQj**).
- Extra experiments in Appendix E-G
- Revised captions for Figure 3, 4, 5, 7, 8 (@reviewer **4fqZ**).
- Fixed a few typos

We hope our pointwise responses below could clarify all reviewers’ confusion. Please kindly let us know if you have any further questions. We will be more than happy to address them fully.

Thanks again for all the reviewers' time.

---

### Meta-Review · Area_Chair_uyU4 · 2023-12-09

**Metareview:**

The paper introduces a framework called CLEAR (Concept-Learning-Enabled metAcognitive inteRvention) designed to enhance the reliability of Large Language Models (LLMs) by enabling self-identification and correction of errors during deployment. The framework is inspired by human cognition and uses a Mixture of Experts (MoE) approach.

While the paper is generally well-presented and easy to understand, reviewers raise concerns about the novelty of the approach, its applicability to other LLM models, and the lack of discussion on computational overhead and scalability. Additionally, questions are raised about potential overfitting and generalization across diverse domains. Reviewers appreciate the effort but suggest addressing these concerns for the paper to be considered for acceptance. The majority of the reviewers were not excited about this paper and this leads to the rejection of this submission.

**Justification For Why Not Higher Score:**

While the paper is generally well-presented and easy to understand, reviewers (including me) raise concerns about the novelty of the approach

**Justification For Why Not Lower Score:**

N/A

---

### Decision · Program_Chairs · 2024-01-16

Reject